# Cephalotin Versus Dicloxacillin for the Treatment of Methicillin-Susceptible *Staphylococcus aureus* Bacteraemia: A Retrospective Cohort Study

**DOI:** 10.3390/antibiotics13020176

**Published:** 2024-02-10

**Authors:** Alejandro Quiñonez-Flores, Bernardo A. Martinez-Guerra, Carla M. Román-Montes, Karla M. Tamez-Torres, María F. González-Lara, Alfredo Ponce-de-León, Sandra Rajme-López

**Affiliations:** 1Internal Medicine Department, Instituto Nacional de Ciencias Médicas y Nutrición Salvador Zubirán, Mexico City 14080, Mexico; quinonez.md95@gmail.com; 2Infectious Diseases Department, Instituto Nacional de Ciencias Médicas y Nutrición Salvador Zubirán, Mexico City 14080, Mexico; bernardo.martinezg@incmnsz.mx (B.A.M.-G.); carla.romanm@incmnsz.mx (C.M.R.-M.); karla.tamezt@incmnsz.mx (K.M.T.-T.); fergonla@gmail.com (M.F.G.-L.); alf.poncedeleon@gmail.com (A.P.-d.-L.); 3Clinical Microbiology Laboratory, Instituto Nacional de Ciencias Médicas y Nutrición Salvador Zubirán, Mexico City 14080, Mexico

**Keywords:** cephalotin, cephalothin, dicloxacillin, Staphylococcus aureus, MSSA, bloodstream infection, bacteraemia

## Abstract

Background: First-line treatments for methicillin-susceptible *S. aureus* (MSSA) bacteraemia are nafcillin, oxacillin, or cefazolin. Regional shortages of these antibiotics force clinicians to use other options like dicloxacillin and cephalotin. This study aims to describe and compare the safety and efficacy of cephalotin and dicloxacillin for the treatment of MSSA bacteraemia. Methods: This retrospective study was conducted in a referral centre in Mexico City. We identified MSSA isolates in blood cultures from 1 January 2012 to 31 December 2022. Patients ≥ 18 years of age, with a first episode of MSSA bacteraemia, who received cephalotin or dicloxacillin as the definitive antibiotic treatment, were included. The primary outcome was in-hospital all-cause mortality. Results: We included 202 patients, of which 48% (97/202) received cephalotin as the definitive therapy and 52% (105/202) received dicloxacillin. In-hospital all-cause mortality was 20.7% (42/202). There were no differences in all-cause in-hospital mortality between patients receiving cephalotin or dicloxacillin (20% vs. 21%, *p* = 0.43), nor in 30-day all-cause mortality (14% vs. 18%, *p* = 0.57) or 90-day all-cause mortality (24% vs. 22%, *p* = 0.82). No severe adverse reactions were associated with either antibiotic. Conclusions: Cephalotin and dicloxacillin were equally effective for treating MSSA bacteraemia, and both showed an adequate safety profile.

## 1. Introduction

*Staphylococcus aureus* is a major cause of infection-related morbidity and mortality, accounting for more than 1 million deaths per year worldwide [1]. Infections caused by *S. aureus* are often complicated by bacteraemia. The mortality rate of methicillin-susceptible *S. aureus* (MSSA) bacteraemia has not significantly improved over time. In 1991 26% of patients with MSSA bacteraemia died [2], compared with 20% in 2009 [3], and 23.7% in 2019 [4]. Penicillinase-stable penicillins such as nafcillin and oxacillin are the treatment of choice for MSSA bacteraemia, with cefazolin used as an alternative [5,6].

Dicloxacillin and nafcillin have similar pharmacokinetic properties, but nafcillin can only be used intravenously while dicloxacillin is well absorbed after oral and intramuscular administration [7,8,9,10]. Cefazolin has some advantages over cephalotin after intravenous administration. Cefazolin’s peak serum concentration is higher, its half-life is greater, and its renal clearance is lower [11,12]. Once administered, cefazolin remains unchanged, while cephalotin is metabolized to its approximately 80% less microbiologically active desacetyl form [11,13].

In Mexico, 47–63% of *S. aureus* isolates are methicillin-susceptible [14]. In ours, like in many other countries, cephalotin is the most widely available first-generation cephalosporin, and dicloxacillin is the only available parenteral anti-staphylococcal penicillin. Lack of access to cefazolin, nafcillin, and oxacillin has driven clinicians to use cephalotin and dicloxacillin to treat MSSA bacteraemia. However, data regarding the safety and efficacy of these antibiotics in MSSA infections is scarce. This study aimed to describe and compare the safety and efficacy of cephalotin and dicloxacillin for the treatment of MSSA bacteraemia.

## 2. Results

We identified 324 MSSA-positive blood cultures in 315 patients. Of these, 113 patients were excluded. The reasons for exclusion were the use of antibiotics other than cephalotin or dicloxacillin (55/113), transfer to other hospitals (28/113), death within 72 h after admission (18/113), incomplete data (7/113), and age <18 (5/113) (Figure 1). We included 202 patients in the analysis, of which 52% (105/202) received dicloxacillin and 48% (97/202) received cephalotin.

The median age was 50 years (IQR 32–63), 59% were male (119/202), and 92% had at least one comorbidity (185/202). Immunosuppression was present in 21% (43/202), and 13% had a malignant neoplasm (27/202). Intensive care unit (ICU) admission was recorded in 30% (60/202). The need for vasopressor therapy and invasive mechanical ventilation (IMV) was documented in 32% (64/202) and 21% (42/202), respectively. The most commonly used empirical antibiotic was vancomycin; the median time from empirical to definitive antibiotic treatment was 3 days (IQR 2–4). Other clinical and demographic data are shown in Table 1.

CLABSI was the most common source of infection (40%, 81/202). The central catheter was removed in all cases, and all patients had follow-up by an infectious disease specialist. Other sources of infection are shown in Figure 2. Complicated bacteraemia was present in 47% (95/202) of patients. Frequent complications were metastatic infection in 67% (64/95), with the osteoarticular system (35/64), and the lungs (21/64) being the most frequent sites of dissemination.

In-hospital all-cause mortality was 21% (42/202), 30-day all-cause mortality was 16.3% (33/202), and 90-day all-cause mortality was 22.7% (46/202). There were no differences between patients receiving cephalotin or dicloxacillin as the definitive treatment (in-hospital mortality 20% vs. 21%, *p* = 0.43; 30-day all-cause mortality 14% vs. 18%, *p* = 0.57; 90-day all-cause mortality 24% vs. 22%, *p* = 0.82). Figure 3 shows the stratified Kaplan–Meier survival estimates.

Univariate logistic regression analysis identified liver cirrhosis (odds ratio [OR] 16.35, 95% confidence interval [CI] 4.26–62.78), ICU admission (OR 6.93, 95% CI 3.31–14-49), vasopressor therapy (OR 10.85, 95% CI 4.93–28.83), need for mechanical ventilation (OR 4.92, 95% CI 2.32–10.43), complicated bacteraemia (OR 2.5, 95% CI 1.24–5.06), and endocarditis (OR 1.93, 95% CI 2.66–26.86) as associated with in-hospital mortality. In the multivariate analysis cirrhosis (aOR 24.7, 95% CI 4.27–143.97), vasopressor therapy (aOR 8.96, 95% CI 2.72–35.37), and endocarditis (aOR 5.34, 95% CI 1.06–26.89) were associated with mortality (Table 2). Similar results were observed when the analysis was performed for 30-day and 90-day mortality.

The median 24 h daily doses of cephalotin and dicloxacillin were 6 g (IQR 2–8) and 8 g (IQR 8–12), respectively. The total daily dose was adjusted to renal function in 44% of patients in the cephalotin group. In patients receiving renal replacement therapy, the median 24 h daily dose of cephalotin was 2 g (IQR, 2–2). Regarding the primary outcome, no differences were observed between those receiving the cephalotin regular dose and the renal-adjusted dose.

There were no severe adverse events recorded for patients in both groups. One patient in each group developed rash during treatment, and one patient in the dicloxacillin group developed non-severe nephrotoxicity.

## 3. Discussion

Cephalotin and dicloxacillin were used due to the lack of access to cefazolin and nafcillin or oxacillin in Mexico. The in-hospital mortality in our study was similar to other reports [15,16]. We found no difference in the primary outcome between patients treated with cephalotin compared to those treated with dicloxacillin. Even though we did not compare cephalotin with cefazolin, and dicloxacillin with oxacillin, their clinical efficacy in MSSA bacteraemia appears to be similar.

Baseline demographics were generally similar between both groups. There was a higher prevalence of COVID-19 in the cephalotin group due to a national shortage of intravenous dicloxacillin during the COVID-19 pandemic. More patients in the dicloxacillin group had autoimmune diseases (34 vs. 17, *p* = 0.02).

In 2015 Rao, S. et al. compared cefazolin vs. oxacillin in 161 patients with MSSA bacteraemia. The primary outcome was treatment failure, defined as the change in MSSA-directed therapy, persistent bacteraemia without resolution of clinical symptoms, recurrent MSSA infection post-negative culture, growth of MSSA from alternative sites after initiation of therapy, or in-hospital mortality secondary to complications from infection. Similar to our study, they did not find statistically significant differences in the primary outcome (cefazolin 15% vs. oxacillin 20%, *p* = 0.72) or in-hospital mortality (cefazolin 1% vs. oxacillin 5%, *p* = 0.23) [17].

Rasmussen et al. retrospectively compared 691 patients with MSSA bacteraemia who received dicloxacillin (368/691) or cefuroxime (323/691), as definitive treatment. The overall 30-day mortality was 20%. They found no difference in 30-day mortality in patients who received dicloxacillin compared to those who received cefuroxime (17.1% vs. 22.9%, *p* = 0.057), but there was a statistically significant difference in 90-day mortality between both groups (dicloxacillin 24.7% vs. cefuroxime 37.5%, *p* = 0.0003) [18]. This difference between groups could be explained by baseline characteristics (there was a higher proportion of patients with severe infection in the cefuroxime group) or confounding factors. In our study, infection severity was similar between groups, and we did not find differences in 30-day or 90-day mortality.

Data comparing cefazolin against nafcillin or oxacillin for the treatment of MSSA bacteraemia have reported similar findings to the ones in our study regarding in-hospital [17], 30-day [19,20], and 90-day mortality [21,22,23,24,25,26].

Recent meta-analyses of retrospective and prospective data comparing anti-staphylococcal penicillins versus cefazolin for the treatment of MSSA bacteraemia have shown similar efficacy of cefazolin and a better safety profile [27,28,29]. The largest prospective study to date, compared 3167 patients with MSSA bacteraemia, who received definitive treatment with cefazolin, nafcillin or oxacillin. These studies report a 37% reduction in 30-day mortality (hazard ratio [HR], 0.63; 95% confidence interval [CI], 0.51–0.78), a 23% reduction in 90-day mortality (HR, 0.77; 95% CI, 0.66–0.90) for patients treated with cefazolin and similar odds for recurrent infection (OR, 1.13; 95% CI, 0.94–1.36) [24].

Because of the chronicity of some clinical syndromes associated with *S. aureus* infection, most studies have reported 90-day mortality. In our study, we decided to report in-hospital mortality as the primary outcome. We considered this to allow us to better evaluate the effect of each antibiotic on infection-associated mortality. Second, patients who received cephalotin and were discharged to continue oral treatment were often treated with other orally available antibiotics.

One concern about cephalosporins for the treatment of MSSA bacteraemia is the inoculum effect. In 2018, Miller et al. evaluated the cefazolin inoculum effect on 89 MSSA isolates from 77 patients with MSSA bacteraemia, from three hospitals in Argentina. They found that 54% of patients harboured isolates that exhibited the inoculum effect, and it was associated with 30-day mortality (39.5% vs. 15.2%, *p* = 0.034) [30]. To describe the cephalotin inoculum effect, Nannini et al. evaluated 98 MSSA isolates of known beta-lactamase (*Bla*) type. They found a modest but significantly higher MIC for the *Bla*-producing strains versus the Bla-non-producing ones. Almost 13% of the strains showed an increase in MIC (8 μg/mL) with the high inoculum, most of which were type B and C Bla-producing strains [31]. In our centre Tovar-Vargas evaluated the cephalotin inoculum effect on 171 MSSA isolates. The MIC_50_ was 0.25 μg/mL with the standard inoculum and 0.5 μg/mL for the high inoculum, concluding that for cephalotin there is not inoculum effect [32].

MSSA bacteraemia is a serious and potentially deadly condition. Our study replicates these observations since 16% and 23% of patients had died within 30 and 90 days after the initial diagnosis, respectively. Moreover, 72% of deaths in our study occurred in the first 30 days after the diagnosis. This confirms other authors’ observations regarding the acute mortality of MSSA bacteraemia [21,22,23,26].

Limitations to our study include the retrospective nature, the non-randomized allocation of treatments, and the lack of a standardized formula for adjusting cephalotin dose to renal function. Despite these limitations, to our knowledge, this is the first study comparing the efficacy of cephalotin versus dicloxacillin for the treatment of MSSA bacteraemia. Some strengths of this study are the inclusion of a diverse population with immunosuppressed and haemodialysis patients, evaluation of mortality both in-hospital and after 30 and 90 days of diagnosis, meticulous documentation of metastatic spread and complications, and consideration of cases during the COVID-19 pandemic.

## 4. Methods

### 4.1. Study Design and Population

This retrospective study was conducted at a tertiary care centre in Mexico City. We identified MSSA isolates in blood cultures from 1 January 2012 to 31 December 2022. Demographic, clinical, laboratory, and radiologic findings were obtained from medical records.

We included adults (≥18 years of age), with a first episode of MSSA bacteraemia, who received cephalotin or dicloxacillin as the definitive antibiotic treatment. We excluded the cases of patients who died in the first 72 h after admission, who were transferred to other units for the treatment of MSSA bacteraemia, or who had a polymicrobial infection. The primary outcome was all-cause in-hospital mortality. Secondary outcomes were all-cause 30-day and 90-day mortality, and adverse events associated with both antibiotics.

Infections were classified as nosocomial, healthcare-associated, and community-acquired infections based on standard definitions [33]. Sources of infection were classified as central line-associated bloodstream infection (CLABSI) [34], skin and soft tissue infection (SSTI), pneumonia, osteoarticular disease, or peritoneal dialysis catheter-related infection, according to the site, organ, or tissue with the highest probability to be the primary source as evaluated by the investigators [35]. Cases were considered as primary bacteraemia when no primary source of infection was identified. Metastatic infection was defined as new foci of infection in sterile sites or organs, not recorded during the initial evaluation [18]. We considered bacteraemia as complicated if the patient had at least one of the following at diagnosis [36,37,38]: endocarditis, implanted mechanical or biological prostheses, MSSA-positive follow-up cultures obtained 48 h after the start of anti-staphylococcal antibiotic treatment, no defervescence after 72 h after anti-staphylococcal antibiotic, and metastatic sites of infection. Immunosuppression was considered when a patient was receiving chemotherapy, immunotherapy, calcineurin inhibitors, mTOR inhibitors, or glucocorticoids (prednisone > 10 mg/kg/day or its equivalent for the previous 2 weeks), at any moment in the 6 months preceding the index bacteraemia. Definitive treatment was considered as the antibiotic used after microbiological identification and susceptibility testing results were available and until the duration of the treatment was completed according to the treating physician.

According to the FDA, serious adverse events were defined as undesirable experiences associated with the use of cephalotin or dicloxacillin resulting in death, life-threatening conditions, prolonged hospitalization, permanent damage, or requiring intervention to prevent permanent damage.

### 4.2. Microbiological Assays

Bacterial isolates were identified using VITEK-2^®^ (bioMérieux, Marcy-lÉtoile, France) and/or MALDI-TOF Biotyper (Bruker, Boston, MA, USA). Susceptibility testing was performed with VITEK-2. Methicillin-susceptible isolates were considered when the VITEK-2 GP ID card showed an oxacillin minimum inhibitory concentration (MIC) ≤ 2 μg/mL and a negative cefoxitin test was registered.

### 4.3. Statistical Analysis

Descriptive statistics were used to summarize patient demographics and clinical characteristics. Data are presented as percentages, means and standard deviations (SD), and medians and interquartile ranges (IQR), as appropriate. Comparisons between groups were made by Fisher’s exact test or Wilcoxon rank sum test, as appropriate. Univariate time-to-event analysis was performed by Kaplan–Meier survival estimates and log-rank tests.

Univariate and multivariate logistic regression analyses for potential risk factors associated with all-cause in-hospital mortality were performed. Variables with a *p*-value < 0.10 in univariate analysis and those with biological plausibility were included in the multivariate model. A two-sided *p*-value ≤ 0.05 was considered statistically significant for all comparisons. All statistical analyses were performed using R version 4.3.1 and RStudio version 2023.03.2.

## 5. Conclusions

The in-hospital, 30-day, and 90-day mortality of monomicrobial MSSA bacteraemia were similar in patients receiving either cephalotin or dicloxacillin. Studies looking into the specific pharmacokinetics of these antibiotics are needed.

## Figures and Tables

**Figure 1 antibiotics-13-00176-f001:**
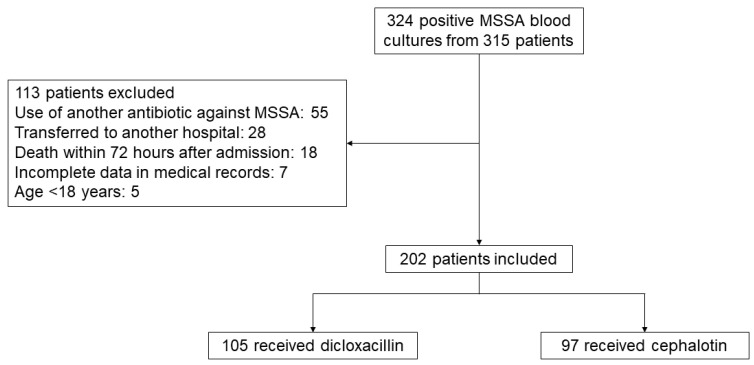
Flow diagram of patient inclusion.

**Figure 2 antibiotics-13-00176-f002:**
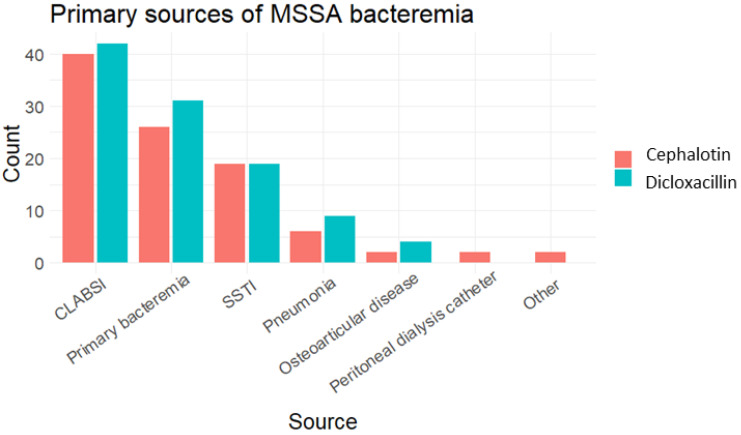
Primary sources of MSSA bacteraemia by frequency, according to treatment group. CLABSI = central line-associated bloodstream infection, SSTI = skin & soft tissue infection.

**Figure 3 antibiotics-13-00176-f003:**
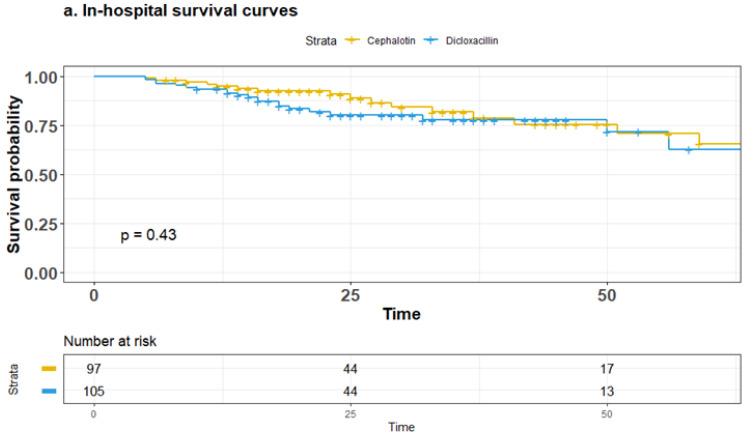
Kaplan–Meier survival curves depicting survival probability for both treatment groups (**a**) In-hospital (**b**) 30-day and (**c**) 90-day.

**Table 1 antibiotics-13-00176-t001:** Baseline clinical characteristics.

	All *n* = 202	Dicloxacillin *n* = 105	Cephalotin *n* = 97	*p*
Age, median (IQR)	50 (32–63)	49 (31–61)	52 (34–65)	0.23
Sex, male, *n* (%)	119 (59)	58 (55)	61 (63)	0.34
BMI, median (IQR)	24 (22–27)	24 (21–27)	24 (22–27)	0.69
Charlson comorbidity index score, median (IQR)	3 (2–4)	2 (1–4)	3 (2–4)	0.15
Any comorbidity, *n* (%)	185 (92)	95 (91)	90 (93)	0.74
COVID-19, *n* (%)	13 (3)	0	13 (13)	<0.001
Diabetes mellitus, *n* (%)	64 (38)	27 (26)	37 (38)	0.08
HbA1c, median (IQR)	7.5 (6.5–8.4)	7.3 (6.2–9.3)	7.7 (7.3–8.4)	
Chronic hypertension, *n* (%)	92 (46)	45 (43)	47 (49)	0.51
End-stage renal disease, *n* (%)	60 (30)	31 (30)	29 (30)	1.00
Hemodialysis, *n* (%)	58 (97)	31 (100)	27 (93)	0.44
Ischemic heart disease, *n* (%)	14 (7)	7 (7)	7 (7)	1.00
Heart failure, *n* (%)	11 (5)	6 (6)	5 (5)	1.00
Heart valve disease, *n* (%)	6 (3)	3 (3)	3 (3)	1.00
Cirrhosis, *n* (%)	13 (6)	7 (7)	6 (6)	1.00
Child-Pugh score A	3 (23)	1	2	
Child-Pugh score B	5 (39)	4	1	0.32
Child-Pugh score C	5 (39)	2	3	
Autoimmune disease, *n* (%)	51 (25)	34 (32)	17 (18)	0.02
HIV infection, *n* (%)	5 (3)	3 (3)	2 (2)	1.00
CD4 cell count, median (IQR)	376 (313–471)	313 (163–510)	423 (400–447)	
Intravenous drug user, *n* (%)	0	0	0	
Solid organ transplant, *n* (%)	21 (10)	11 (11)	10 (10)	1.00
Renal transplant, *n* (%)	21	11	10	
Bone marrow transplant, *n* (%)	1 (0.5)	0	1 (1)	0.97
Autologous, *n* (%)	1		1	
Malignant neoplasm, *n* (%)	26 (13)	11 (11)	15 (16)	0.40
Solid organ neoplasm, *n* (%)	10	3	7	
Hematologic malignancy, *n* (%)	16	8	8	
Pharmacologic immunosuppression, *n* (%)	43 (21)	26 (25)	17 (18)	0.28
ICU, *n* (%)	60 (30)	29 (28)	31 (32)	0.60
Vasopressor, *n* (%)	64 (32)	31 (30)	33 (34)	0.59
IMV, *n* (%)	42 (21)	21 (20)	21 (22)	0.91
Hospital-acquired infection, *n* (%)	42 (21)	18 (17)	24 (25)	0.25
Complications, *n* (%)	95 (47)	51 (49)	44 (45)	0.75
Persistently positive blood culture	48 (24)	23 (22)	25 (26)	
Endocarditis	14 (7)	6 (6)	8 (8)	
Infection of implantable device	4 (2)	1 (1)	3 (3)	
Empyema	4 (2)	2 (2)	2 (2)	
Persistent fever	3 (2)	1 (1)	2 (2)	
Distant hematogenous spread, *n* (%)	64 (32)	35 (33)	29 (30)	0.71
Lung	21 (10)	10 (10)	11 (11)	
Joints	15 (7)	7 (7)	8 (8)	
Bone	11 (5)	6 (6)	5 (5)	
Vertebrae	9 (5)	5 (5)	4 (4)	
CNS	8 (4)	4 (4)	4 (4)	
Psoas	6 (3)	3 (3)	3 (3)	
Kidney	1 (1)	1 (1)	0	
Time to definitive antibiotic therapy in days, median (IQR)	3 (2–4)	3 (2–4)	3 (2–4)	0.57
Length of stay in days, median (IQR)	23 (16–36)	21 (16–35)	24 (17–38)	0.26

BMI = body mass index, CNS = central nervous system, HbA1c = glycated haemoglobin, HIV = human immunodeficiency virus, ICU = intensive care unit, IMV = invasive mechanical ventilation, IQR = interquartile range.

**Table 2 antibiotics-13-00176-t002:** Logistic regression univariate and multivariate analysis for all-cause in-hospital mortality.

	Univariate		Multivariate *p* < 0.10	
	aOR (95% CI)	*p*	aOR (95% CI)	*p*
COVID-19	2.57 (0.79–8.30)	0.12		
Diabetes mellitus	1.86 (0.92–3.75)	0.08	1.90 (0.70–5.20)	0.21
Chronic hypertension	1.11 (0.56–2.19)	0.76		
End-stage renal disease	0.49 (0.21–1.13)	0.09	0.39 (0.11–1.34)	0.13
Ischemic heart disease	1.04 (0.28–3.92)	0.95		
Heart failure	0.84 (0.17–4.04)	0.83		
Cirrhosis	16.35 (4.26–62.78)	<0.001	23.84 (4.10–138.52)	<0.001
Autoimmune disease	0.64 (0.28–1.49)	0.30		
Solid organ transplant	0.17 (0.02–1.31)	0.09	0.32 (0.03–4.01)	0.38
Malignant neoplasm	0.66 (0.22–2.03)	0.47		
Pharmacologic immunosuppression	0.23 (0.07–0.79)	0.02	0.36 (0.07–1.76)	0.20
Intensive care unit admission	6.93 (3.31–14.49)	<0.001	3.03 (0.62–14.70)	0.17
Vasopressor	10.85 (4.93–28.83)	<0.001	8.27 (2.14–31.94)	0.002
IMV	4.92 (2.32–10.43)	<0.001	0.46 (0.11–2.00)	0.30
Hospital-acquired infection	1.73 (0.78–3.78)	0.17		
Duration of bacteremia (days)	1.04 (0.94–1.14)	0.47		
Complicated bacteremia	2.50 (1.24–5.06)	0.01	1.47 (0.26–8.45)	0.67
Persistently positive blood culture	2.14 (1.02–4.47)	0.04	1.57 (0.42–5.92)	0.51
Endocarditis	8.46 (2.66–26.86)	<0.001	6.92 (1.39–34.50)	0.02
Persistent fever	1.93 (0.17–21.78)	0.60		
Distant hematogenous spread	1.86 (0.92–3.75)	0.08	0.78 (0.19–3.15)	0.72
Lung	2.66 (1.02–6.92)	0.05		
Joints	0.95 (0.26–3.53)	0.94		
Bone	0.84 (0.17–4.04)	0.83		
Vertebrae	1.09 (0.22–5.47)	0.91		
CNS	0.53 (0.06–4.46)	0.56		
Psoas	1.95 (0.35–11.03)	0.45		
Definitive antibiotic treatment		0.95		0.43
Dicloxacillin	Ref		Ref	
Cephalotin	0.98 (0.49–1.93)		0.69 (0.27–1.74)	

aOR = adjusted odds ratio, CI = confidence interval, CNS = central nervous system, IMV = invasive mechanical ventilation.

## Data Availability

The datasets used and/or analysed during the current study are available from the corresponding author (S.R.-L.) upon reasonable request.

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
