# Peer review of "Cephalotin Versus Dicloxacillin for the Treatment of Methicillin-Susceptible Staphylococcus aureus Bacteraemia: A Retrospective Cohort Study"

_antibiotics, 2024, doi:10.3390/antibiotics13020176_

Round 1
Reviewer 1 Report
Comments and Suggestions for Authors
The layout and logical flow and English Grammar need to be improved. The statistical tables also need work and possible reformatting
Comments on the Quality of English LanguageThis is highly useful knowledge, however the presentation needs to improve, especially in regard to the layout of the tables and figures. The explanation of the findings also needs revision... it is not clear.
Author Response
Reviewer 1
Comments and Suggestions for Authors
The layout and logical flow and English Grammar need to be improved. The statistical tables also need work and possible reformatting
Response: All tables have been reviewed and some terms have been changed to more frequently used ones.
Comments on the Quality of English Language
This is highly useful knowledge, however the presentation needs to improve, especially in regard to the layout of the tables and figures. The explanation of the findings also needs revision... it is not clear.
Response: Tables and Figures have been revised and updated accordingly. The phrasing of the results section has been modified to be clearer.
Reviewer 2 Report
Comments and Suggestions for Authors
Authors describe and compare the safety and efficacy of cephalotin and dicloxacillin for the treatment of MSSA bacteraemia.
Authors should be format references and tables as request by the journal.
Introduction: I suggest to improve this section. In particular, authors should be introduce the reason of which they used cephalotin and not cefazolin. In my opinion this is very important to give strength to the follow-up of the manuscript, also since the discussion is mainly based on studies that compare the cefazolin efficacy.
Methods:
I suggest to divide this section in three parts:
- study design and population
- microbiological assays
- statistical analysis
lane 53: "This retrospective was study conducted at a tertiary": improve the English form
lane 85-87: in my opinion not is the right location for this sentence
Results:
I suggest to move the figure 1 after lane 111
Authors should be better explain figures 2 and 3 in the caption.
Discussion:
Also in this section I would explain why the cephalotin was chosen instead of the cefazolin
Author Response
Reviewer 2
Comments and Suggestions for Authors
Authors describe and compare the safety and efficacy of cephalotin and dicloxacillin for the treatment of MSSA bacteraemia.
Authors should be format references and tables as request by the journal.
Response: references and tables are now in the requested format.
Introduction: I suggest to improve this section. In particular, authors should be introduce the reason of which they used cephalotin and not cefazolin. In my opinion this is very important to give strength to the follow-up of the manuscript, also since the discussion is mainly based on studies that compare the cefazolin efficacy.
Response: We have added a sentence stating that “cefazolin is the most widely available first-generation cephalosporin, and dicloxacillin is the only available parenteral anti-staphylococcal penicillin”
Methods:
I suggest to divide this section in three parts:
- study design and population
- microbiological assays
- statistical analysis
Response: We are thankful for this suggestion and have divided the methods section accordingly.
lane 53: "This retrospective was study conducted at a tertiary": improve the English form
Response: The line now reads “This retrospective study was conducted at…”
lane 85-87: in my opinion not is the right location for this sentence
Response: We have relocated that information to the subsection Study design and population
Results:
I suggest to move the figure 1 after lane 111
Response: We agree and have moved Figure 1 accordingly.
Authors should be better explain figures 2 and 3 in the caption.
Response: The captions for both figures now include more information.
Discussion:
Also in this section I would explain why the cephalotin was chosen instead of the cefazolin
Response: We have added an explanation regarding the lack of acces to cefazolin and nafcillin in Mexico.
Reviewer 3 Report
Comments and Suggestions for Authors
The manuscript entitled “Cephalotin versus dicloxacillin for the treatment of methicillin-susceptible Staphylococcus aureus bacteraemia: a retrospective cohort study” is interesting, well-written and addresses a relevant issue.
Methicillin-susceptible Staphylococcus aureus (MSSA) bacteremia is indeed a serious and potentially deadly disease. Therefore, studies such as the one presented here, which analyses the possibility of using alternative antibiotics in therapy, are important.
The study was well conducted, the results are adequately treated, and are presented and discussed in a clear, logical, and coherent way. The bibliographical references are adequate.
Therefore, the manuscript should be considered for publication after some minor corrections/revisions are made.

Author Response
Reviewer 3
Comments and Suggestions for Authors
The manuscript entitled “Cephalotin versus dicloxacillin for the treatment of methicillin-susceptible Staphylococcus aureus bacteraemia: a retrospective cohort study” is interesting, well-written and addresses a relevant issue.
Methicillin-susceptible Staphylococcus aureus (MSSA) bacteremia is indeed a serious and potentially deadly disease. Therefore, studies such as the one presented here, which analyses the possibility of using alternative antibiotics in therapy, are important.
The study was well conducted, the results are adequately treated, and are presented and discussed in a clear, logical, and coherent way. The bibliographical references are adequate.
Therefore, the manuscript should be considered for publication after some minor corrections/revisions are made.
- Methods
(line 92) – the subheading "2.2 Statistical analysis" should be moved to the
next page
Response: in the new version, this subheading is correctly situated
- Results
(lines 106-111) – At the beginning of the paragraph (2nd sentence) it is
stated that: "Of these, 315 patients were screened and 113 were excluded."
The reasons that led to the exclusion of 101 of the 113 patients are then
detailed: 55 due to the use of other antibiotics, 28 because they had been
transferred to other hospitals and 18 for death within 72 hours of hospital
admission. I believe that the information that led to the exclusion of the 12
missing patients should also be included (7 patients due to incomplete
medical records + 5 due to being minors).
Response: we have included this information in the results section.
(line 109) - "(Error! Reference source not found.)." appears at the end of the
sentence; I assume you meant to include the indication (strictly necessary) to
see Figure 1; please correct.
Response: This error message has been corrected and now reads “Figure 1”
(line 119) - a period is missing at the end of the sentence.
Response: We have added the missing period.
(line 128) – again appears: "Error! Reference source not found", at the end
of the sentence; I believe the aim was to direct the reader to the figure
showing the primary sources of MSSA bacteraemia; in the text this figure is
numbered "3", which is not correct; the figure should be renumbered "Figur
2" and included as close as possible to this paragraph (certainly before the
figure referring to the stratified Kaplan Meier survival estimates).
Response: We are thankful for these observations. Figures have been renumbered and relocated. The error message has been corrected and now reads “Figure 2”
(lines 129-130) – the sentence "Frequent complications were metastatic
infection in 67% had a metastatic infection" is not well written (redundant,
repeated content). Please amend this aspect.
Response: This mistake has been corrected. The sentence now reads “Frequent complications were metastatic infection in 67%...”
(line 133) - Figure 2 should be renumbered to Figure 3; this way it will be in
accordance with the content of the text and also with the way it is
referenced later on line 139
Response: We are thankful for these observations. Figures have been renumbered and relocated.
(line 141) - for the sake of simplicity in the interpretation of figure 3, I
believe that a legend should be included with the meaning of CLABSI and
SSTI
Response: We have added the meanings of both abbreviations in the Figure caption.
Round 2
Reviewer 1 Report
Comments and Suggestions for Authors
Much improved. Better linking of research
question with research findings. There was no
difference in mortality. The cause of death was not S.aureus bacterial sepsis so presumably both antibiotics were equally effective.
Author Response
Much improved. Better linking of research question with research findings. There was no difference in mortality. The cause of death was not S.aureus bacterial sepsis so presumably both antibiotics were equally effective.
Response. We are thankful for the second revision of the manuscript. Both antibiotics were indeed equally effective, as stated in the manuscript.
Reviewer 2 Report
Comments and Suggestions for Authors
I thank the authors for improving their manuscript, which I suggest for publication in the Antibiotics journal.
Author Response
I thank the authors for improving their manuscript, which I suggest for publication in the Antibiotics journal.
Response. We are thankful for the second review of the manuscript.
Reviewer 3 Report
Comments and Suggestions for Authors
The revised version includes the suggested corrections and has the quality required for publication in the journal.
Author Response
The revised version includes the suggested corrections and has the quality required for publication in the journal.
Response. We are thankful for the second revision of the manuscript.